# Research on the Path Planning of Unmanned Sweepers Based on a Fusion Algorithm

**Yongjie Ma** [1]**, Peng Ping** [1,2] **and Quan Shi** [1,2,*]

1    School of Information Science and Technology, Nantong University, Nantong 226000, China
2    School of Transportation and Civil Engineering, Nantong University, Nantong 226000, China
*    Correspondence: sq@ntu.edu.cn

**Abstract:** Path planning is one of the key technologies for unmanned driving. However, global paths are unable to avoid unknown obstacles, while local paths tend to fall into local optimality. To solve the problem of unsmooth and inefficient paths on multi-angle roads in a park which cannot avoid unknown obstacles, we designed a new fusion algorithm based on the improved A* and Open_Planner algorithms (A-OP). In order to make the global route smoother and more efficient, we first extracted the key points of the A* algorithm and improved the node search structure using heap sorting, and then improved the smoothness of the path using the minimum snap method; secondly, we extracted the key points of the A* algorithm as intermediate nodes in the planning of the Open_Planner algorithm, and used the A-OP algorithm to implement the path planning of the unmanned sweeper. The simulation results show that the improved A* algorithm significantly improved the planning efficiency, the nodes are less computed and the path is smoother. The fused A-OP algorithm not only accomplished global planning effectively, but also avoided unknown obstacles in the path.

**Keywords:** A* algorithm; heap sorting; Open_Planner algorithm; fusion algorithm; unmanned sweeper





## 1. Introduction

With the development of urban transportation, the emergence of driverless sweepers has greatly changed the way cities are cleaned. Replacing the traditional manual road-cleaning model with driverless [1,2] sweeping not only improves cleaning efficiency, but also reduces labor costs. An important prerequisite for driverless sweepers to drive autonomously is to plan an optimal and safely executable path. The principle of path planning is to plan a path from the starting point to the end point based on regional environmental information and vehicle information, and it also needs to have an obstacle-avoidance function during the driving process.

Path-planning algorithms can be divided into two categories, one is global path planning and the other is local path planning. Global path planning is a static planning algorithm, which plans an optimal route from the start point to the end point on the map based on the existing map environment information. Global-planning algorithms include classical algorithms such as the A* algorithm [3,4], Dijkstra algorithm [5,6], RRT algorithm [7,8], and also intelligent algorithms such as the ant colony algorithm [9,10] and genetic algorithm [11,12]. Local path planning is implemented using a dynamic-planning algorithm [13,14], which is based on a sensor sensing the surrounding environment to plan a path for the vehicle to drive safely, and is often applied to scenarios such as overtaking and obstacle avoidance [15]. Local-path-planning algorithms include the dynamic window method, artificial potential field method [16,17], Open_Planner algorithm [18], Bessel curve algorithm [19,20] and so on. There are also newly proposed neural network algorithms [21,22], and others.

The dominant planning algorithm used by many unmanned-vehicle planning frameworks is the A* algorithm, such as the move_base framework [23], and the Autoware framework [24]. For example, Dijkstra's breadth search algorithm can complete the pathfinding operation, but it has the disadvantage of consuming a large amount of computer memory, being computationally intensive with many nodes, and having a high space and time complexity. It is, therefore, less often used for the actual path planning of unmanned vehicles. However, traditional A* algorithms suffer from the problem of low algorithm efficiency due to many redundant points, on the one hand, and low path smoothing in the case of many turning points, on the other hand, resulting in less smooth vehicle turns and extra time or power consumption when turning. At the same time, the A* algorithm cannot avoid unknown static or dynamic obstacles, so how to make the path smooth, improve planning efficiency and avoid obstacles is a problem that needs to be solved. In reality, the use of global path-planning algorithms alone does not guarantee the safety of an unmanned vehicle, as the presence of unknown obstacles can make the vehicle unsafe.

Many researchers have conducted in-depth research and discussions on planning algorithms. Zhao [25] proposed a Maklink-based approach to address the shortcomings of the A* algorithm in terms of its computationally intensity and limitations of domain search strategies. Wang [26] designed a method which introduced extended distance, bidirectional searching and smoothness into path planning to improve paths by reducing the number of turns. Xiong [27] proposed a conventional algorithm which does not consider the vehicle model and lacks speed planning. The safe-space configuration of the vehicle model was considered and a vehicle acceleration system was constructed to improve the applicability of the A* algorithm in vehicle planning. Li [28] combined a time-factor normalization model with a valuation function to reduce the cost of trip planning. Droge Greg [29] proposed a two-model predictive control framework which incorporates information from path planning into a dynamic windowing method for obstacle avoidance, ensuring that target locations are detected in an unknown environment. Zhang [30] used a hopping point search algorithm to extend the hopping points on sub-nodes and improve the efficiency of the A* algorithm. Zhang [31] proposed a bidirectional A* algorithm based on third-order Bessel curve trajectory optimization, which solved the problem of multiple folds and large corners in the search path. Y. Bian [32] defined a new evaluation sub-function to obtain a new DWA evaluation function, which solved the problem of falling into evaluation function failure when the mobile robot encountered an obstacle. The planning algorithms chosen in this paper are the improved A* algorithm and the Open_Planner algorithm. Both the traditional A* algorithm and the Open_Planner algorithm can be applied to the path-planning problem. However, the disadvantages of the traditional algorithm, the A* algorithm, are reduced efficiency due to the presence of a large number of non-essential nodes, unsmooth routes around corners and difficulty in dealing with unknown obstacles when searching in complex environments. Open_Planner requires a global environment to establish the best locally planned paths. To address these issues, we fuse the global-path-planning algorithm with the local-path-planning algorithm. To improve the planning efficiency of the A* algorithm, we extract key points based on obstacles and map boundaries in the A* algorithm. The slowest part of the A* algorithm is finding the node with the smallest F value in the open list, and repeatedly searching large lists slows down the process. Therefore, we use a binomial heap to optimize the way the A* algorithm stores the list, placing the smallest points at the top of the heap to improve the algorithm's search efficiency. Unsmooth paths around corners can also have an effect on vehicle operation; thus, we make the paths smoother using the minimum snap method. Afterwards, the key points of the A* algorithm are used as intermediate target points for the Open_Planner algorithm and the two algorithms are fused. With the planning of the A-OP fusion algorithm, the vehicle has an optimal path and, at the same time, achieves the safety effect of obstacle avoidance.

In summary, we aim to realize a driverless sweeper vehicle which can both drive with the global optimal planning route and reach the destination safely by avoiding the obstacles in real time. In this paper, based on the traditional path-planning algorithm,

the key points of the algorithm are selected and the node search structure is improved to enhance the smoothness of the path, after which the global-planning algorithm and the local-path-planning algorithm are fused by key points and the algorithm is fused. The contribution of the work in this paper is shown in the following aspects:

- A new route-planning method by fusing improved A* and Open_Planner;
- Feature point search strategy for planning paths;
- Path smoothing strategy based on minimum snap.

## 2. Traditional A* Algorithm with Improvements

*Principles of the Traditional A* Algorithm*

The A* algorithm is one of the traditional global-path-planning algorithms applied in various addressing scenarios. A* algorithm is an improvement of Dijkstra's algorithm, which optimizes for a single objective point using a heuristic function, giving priority to the path closer to the objective. Its evaluation function consists of two components, the actual generation value and the estimated generation value, and the definition can be expressed as in Equation (1).

$$F(n) = G(n) + H(n) \tag{1}$$

where $n$ denotes the current node, $G(n)$ denotes the actual generation value from the starting point to node $n$, and $H(n)$ is a heuristic function which denotes the estimated generation value from node $n$ to the target point. If we want to find the optimal solution for the two-point path, the key is the selection of the estimation function $H(n)$. If $H(n)$ is less than or equal to the actual value of the distance from the current node to the target node, then the search nodes are many, the search range is large, and the efficiency is low, but the optimal solution can be obtained. If $H(n)$ is equal to the actual value, then the search will be carried out strictly according to the shortest path; if $H(n)$ is greater than the actual value, the search nodes are fewer, the search range is smaller, and the efficiency is high, but the optimal solution cannot be guaranteed. Therefore, an exact heuristic function is constructed to calculate the length of the shortest path between any two nodes. In two-dimensional maps, calculating the actual distance value between nodes and target points as a surrogate value can achieve a better result. The traditional A* algorithm heuristic functions are usually Euclidean distance, Chebyshev distance and Manhattan. Their distance equations are shown as follows, Equation (2), Equation (3), Equation (4), respectively.

$$\text{Euclidean distance}: \quad d = \sqrt{(x_2 - x_1)^2 + (y_2 - y_1)^2} \tag{2}$$

$$\text{Chebyshev distance}: \quad d = max(|x_1 - x_2|, |y_1 - y_2|) \tag{3}$$

$$\text{Manhattan}: \quad d = |x_1 - x_2| + |y_1 - y_2| \tag{4}$$

Figure 1 shows a graphical representation of the three distance formulas. The traditional A* algorithm generally uses Euclidean distance or Manhattan distance, but Chebyshev distance only needs to calculate addition and subtraction, which greatly improves the efficiency of operation, and there will be no error no matter how many times the superposition is calculated. Therefore, the Chebyshev distance formula was chosen in this paper.

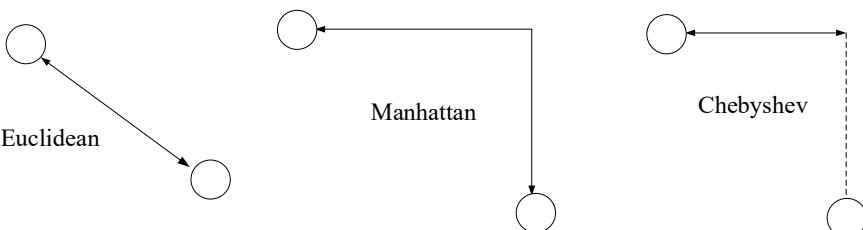

**Figure 1.** Diagram of distance measurement methods.

## 3. Method

In order to solve the problems of many redundant points, uneven paths and low search efficiency in the path planning of A* algorithm, this paper screened and extracted the nodes in the path of A* algorithm, deleted the redundant points in the path, and improved the smoothness of the path. At the same time, heap sort was used to improve the storage structure of algorithm nodes to improve the search efficiency. Unlike a road, a large campus such as a park has a large number of buildings and traffic off-ramps, which results in a large number of turning points. The traditional A* algorithm has the defect of unsmoothed path, which leads to the unmanned vehicle not being silky smooth when driving and turning according to the path planning. At the same time, A* algorithm has the problem of many redundant points in planning, which leads to more time spent in path planning. Therefore, in order to solve these problems, this paper filtered and extracted the nodes in the path of A* algorithm, removed the redundant points in the path, and improved the smoothness of the path using minimum snap. At the same time, heap sorting was used to improve the storage structure of the algorithm nodes and to improve the search efficiency.

### 3.1. Optimized Node Data Structure Based on Binomial Heap

A* algorithm needs to scan all the nodes with the smallest F value in the OpenList table in order. Although it can complete all the node scanning, the speed is obviously slow. Literature [33] proposes that when the number of nodes in the OpenList table reaches a certain value, the earliest and larger nodes should be deleted. Although this method can keep the number stable, if there are many obstacles in a large area, it will consume a lot of time to search each node in order. Heap sort is a kind of selection sort using the nature of heap. It selects the smallest node using the relationship between parent node and child node in binary tree, so that the node with the smallest $F(n)$ value is placed at the top of the list for access, which improves the search efficiency. The idea of heap sort is as follows: the array of nodes in the two tables is resized into a small root heap whose top node is the smallest node in the heap. The top node of the small root heap is switched with the last node of the unordered region, the last node added to the ordered region, the new node order adjusted, and this operation is repeated until the unordered region is empty. Figure 2 shows the process of minimum heap sort.

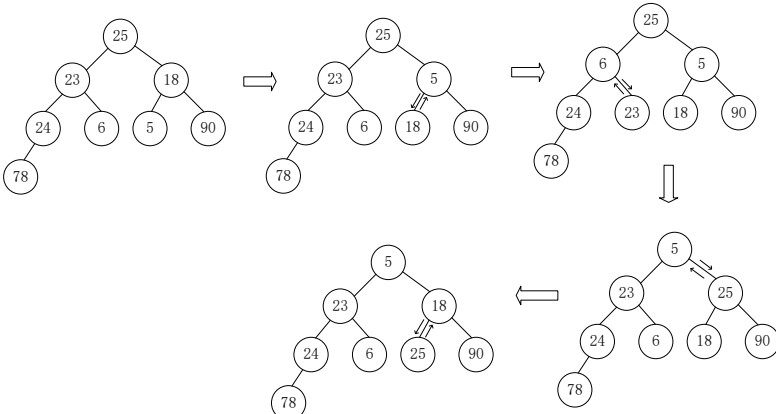

**Figure 2.** Minimum heap sort process.

### 3.2. Extract Key Points Based on Raster Map Boundaries and Obstacles

A* algorithm will produce many redundant points in the addressing process, which will affect the planning efficiency. Therefore, the expansion nodes in path planning were screened and the key points in the path were extracted to improve the algorithm. The redundant nodes in the node expansion process of the A* algorithm were removed, and only key nodes were retained as expansion points for path planning.

In this paper, a large number of redundant points were excluded and the number of unnecessary nodes was reduced by searching the key points in the path based on raster

map boundaries and obstacles. Two key points were connected to realize the jump between two points, so as to improve the efficiency of path search. The key point search process is as follows:

　　　Step 1: In the OpenList table, select a node with the minimum generation value or select a starting node to start the search.

　　　Step 2: Search search direction in the line or oblique search only. The search ends if a node is found, an obstacle is encountered, or a raster map boundary is hit. Add the searched nodes to the OpenList table.

　　　Step 3: If the search is not completed in the oblique direction, step forward in the oblique direction and repeat the above process.

　　　Step 4: If the comprehensive search has been completed, the current node is considered to have been completely searched, and the current node is removed from the OpenList table and added to the ClostList table.

　　　Step 5: Repeat the search for the node with the lowest weight in the ClostList table until the OpenList table is empty or the final point is found. The points stored in the CloseList table are the key points to search for.

　　　As shown in Figure 3, the red grid X is the starting point, the green grid Y is the end point, the blue X5 and X9 are the key points, and the path connected by nodes $X \rightarrow X_5 \rightarrow X_9 \rightarrow Y$ is the final path planned by the algorithm. Compared with the traditional A* algorithm, the improved A* algorithm only leaves the starting point, the ending point, and the two key points $X_5$ and $X_9$; deletes a large number of redundant points; and adds the key points to the OpenList table for cost evaluation and path planning. The improved A* algorithm can make long-distance connections through key points, and it takes less time to reach the target point, which greatly improves the planning efficiency.

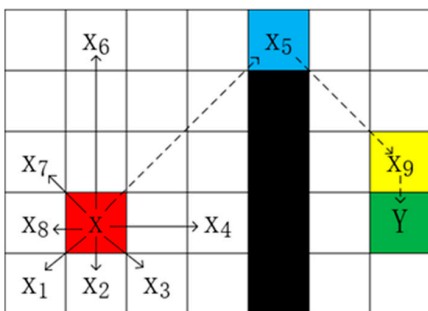

**Figure 3.** Key point extraction diagram. (The red square is the starting point, the blue square and the yellow square are the characteristic points in the path, and the green square is the ending point).

### 3.3. Path Smoothing Based on Minimum Snap

　　　The path searched by A* algorithm does not consider the kinematic model, so there will be unsmooth points at the corner. The robot does not have a motion mutation at one point. If it is a universal wheel robot, it also needs to rotate at a certain angle to continue forward, which obviously wastes a lot of time and efficiency. Therefore, based on the track points in the track, a smooth curve connecting each track point is planned. This paper samples the trajectory smoothing processing based on minimum snap. It is difficult to express a complex trajectory with only one polynomial, so we divide the trajectory into multiple segments according to time, and each segment is expressed using a polynomial curve, each with the following equation:

$$
p(t) = \begin{cases}
[1, t, t^2, \ldots, t^n] \cdot p_1 & t_0 \leq t \leq t_1 \\
[1, t, t^2, \ldots, t^n] \cdot p_2 & t_1 \leq t \leq t_2 \\
[1, t, t^2, \ldots, t^n] \cdot p_3 & t_2 \leq t \leq t_3 \\
\qquad \cdots \\
[1, t, t^2, \ldots, t^n] \cdot p_k & t_{k-1} \leq t \leq t_k
\end{cases}
\tag{5}
$$

$K$ is the number of segments of the trajectory. $p_i = [p_{i0}, p_{i1}, p_{i2}, \ldots, p_{in}]^T$ is the parameter vector of the i-th segment trajectory. We want smoothness (position and velocity, etc.) at adjacent trajectories. Usually, there are many trajectories that satisfy the constraints, but we only need one specific trajectory; therefore we need to construct an optimal function to find the optimal trajectory among the feasible trajectories. Thus, we modelled the problem as a constrained optimization problem, as in Equation (6), below.

$$\begin{cases} \min f(p) \\ s.t. \quad A_{eq} p = b_{eq} \\ \quad A_{ieq} P \leq b_{ieq} \end{cases} \tag{6}$$

In Equation (6), $p$ denotes the trajectory parameter; *s.t.* means constraint, meaning subject to. All we had to do is to list the $A_{eq}$, $b_{eq}$, $A_{ieq}$, bieq parameters from the optimization problem and put them into the optimizer (a library in Matlab) to solve for the trajectory parameter $p$.

The target trajectory parameter $p$ was solved using the optimization method. First, according to the path point, the trajectory was divided into k segments, the distance of each segment calculated, according to the distance equalization time $T$, to obtain the time series $t_0$, $t_1$, $t_2, \ldots, t_k$. The time allocation assigned the total time T to each segment according to the distance of each segment. Then, the optimization function was constructed. The optimization function for minimum snap is shown in Equation (7), Equations (8) and (9) is the solution for $Q_i$.

$$\min \int_0^T \left( P^{(4)}(t) \right)^2 dt = \min \sum_{i-1}^k \int_{t_{i-1}}^{t_i} \left( P^{(4)}(t) \right)^2 dt = \min \sum_{i=1}^k p^T Q_i p \tag{7}$$

$$Q_i = \int_{t_{i-1}}^{t_i} \left[ 0, 0, 0, 24, \ldots, \frac{n!}{(n-4)!} t^{n-4} \right]^T \left[ 0, 0, 0, 24, \ldots, \frac{n!}{(n-4)!} t^{n-4} \right] dt \tag{8}$$

$$Q_i = \begin{bmatrix} 0_{4 \times 4} & 0_{4 \times (n-3)} \\ 0_{(n-3) \times 4} & \frac{r!}{(r-4)!} \frac{c!}{(c-4)!} \frac{1}{(r-4)+(c-4)+1} \left( t_i^{(r+c-7)} - t_{i-1}^{(r+c-7)} \right) \end{bmatrix} \tag{9}$$

where $r$ and $c$ are the row indexes and column indexes of the matrix, with indexes starting from 0 and the first row $r = 0$; $p^{(4)}(t)$ represents the 4th-order derivative with respect to time $t$.

$$Q = \begin{bmatrix} Q1 & 0 & 0 \\ 0 & \ddots & 0 \\ 0 & 0 & Qn \end{bmatrix} \tag{10}$$

For the robot to satisfy the continuity constraint before the intermediate point, the robot's velocity and acceleration at the end of the *j*-1 segment trajectory was set equal to the velocity and acceleration at the initial moment of the *j* segment trajectory. Figure 4 is a schematic diagram of constraint conditions. The yellow circle is the node, and the curve is the track between two nodes. The expression is shown as follows:

$$f_j^{(k)}(T_j) = f_{j+1}^{(k)}(T_j) \tag{11}$$

Simplifying the above equation yields the following equation. Equation (12) represents the continuity constraint between the two segments.

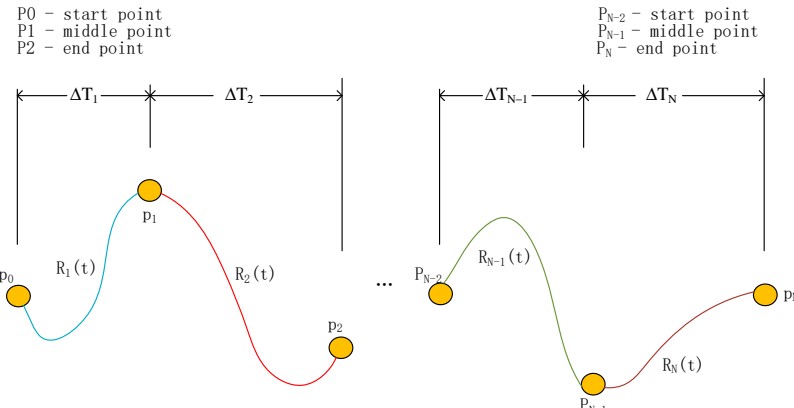

**Figure 4.** Schematic diagram of the continuity constraint.

$$f_j^{(k)}(T_j) = f_{j+1}^{(k)}(T_j)$$

$$\sum_{i \geq k} \frac{i!}{(i-k)!} T_j^{i-k} p_{i,j} - \sum_{i \geq k} \frac{l!}{(l-k)!} T_j^{l-k} p_{j+1,i} = 0$$

$$\left[ \cdots \frac{i!}{(i-k)!} T_j^{i-k} \cdots - \frac{l!}{(l-k)!} T_j^{l-k} \cdots \right] \begin{bmatrix} \vdots \\ p_{j,i} \\ \vdots \\ p_{j+1,i} \\ \vdots \end{bmatrix} = 0 \qquad (12)$$

$$\begin{bmatrix} A_j - A_{j+1} \end{bmatrix} \begin{bmatrix} P_j \\ P_{j+1} \end{bmatrix} = 0$$

The final constraint obtained is shown in Equation (13)

$$A_{eq} \begin{bmatrix} p_1 \\ \vdots \\ p_M \end{bmatrix} = d_{eq} \qquad (13)$$

Combining the evaluation function and the constraints leads to Equation (13). Finally, the final trajectory is obtained by bringing the values of the coordinates of the start, end and feature points, as well as the values of time, velocity and acceleration into Equation (14).

$$p^T Q_i p = min \begin{bmatrix} P_1 \\ \vdots \\ P_M \end{bmatrix}^T \begin{bmatrix} Q_1 & 0 & 0 \\ 0 & \ddots & 0 \\ 0 & 0 & Q_M \end{bmatrix} \begin{bmatrix} P_1 \\ \vdots \\ P_M \end{bmatrix}$$

$$s.t. \ A_{eq} \begin{bmatrix} P_1 \\ \vdots \\ P_M \end{bmatrix} = d_{eq} \qquad (14)$$

## 4. Open_Planner Algorithm

The optimal path not only needs global planning, but also needs local dynamic programming to avoid obstacles and ensure safety. In this paper, Open_Planner algorithm was used for local path planning.

### *The Storage Structure of Nodes Is Optimized*

The Open_Planner algorithm uses roll-outs generator to obtain multiple local sampling trajectories according to the path planned by the whole route, and calculates the normalized

cost function of each sampling trajectory through obstacle detection and cost function toll-outs evaluator. The trajectory line with the lowest cost and the best is used as the local path.

The Open_Planner partial plan is divided into two parts: roll-outs generator and roll-outs evaluator. The roll-outs generator is used to explore local paths and optimize trajectories, and the roll-outs evaluator is used to select the optimal path for the cost function.

The input information of the roll-outs generator is the current location of the vehicle, the global planning path, and rollouts. Rollouts are divided into three parts: car tip, roll-in, and roll-out, as shown in Figure 5.

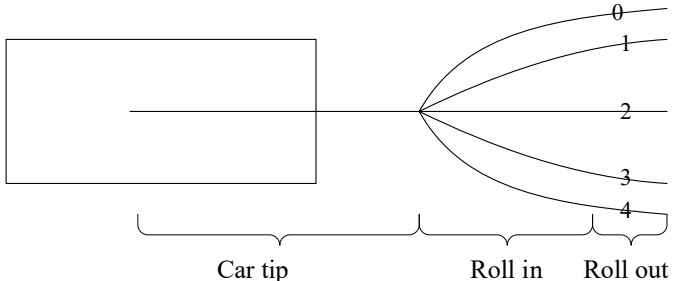

**Figure 5.** Components of rollouts.

Car tip represents the starting point from the center of the vehicle to the horizontal sampling, 'Roll-in' represents the starting point from the horizontal sampling to the parallel sampling, and 'Roll-out' represents the maximum planning distance from the starting point of the parallel sampling. By intercepting the global path, point sampling is carried out for the captured global path, and the final trajectory is generated by smoothing the obtained sampling points.

The roll-outs generator first extracts car tip, roll-in and roll-out from the global path according to the current position of the vehicle, then conducts transverse sampling for the extracted global path, and, finally, makes the path smooth using conjugate gradient.

Roll-outs evaluator evaluates different paths using three cost functions, namely, priority cost, collision cost, and transition cost. Priority cost indicates that the middle path is preferred when there is no obstacle. There are two types of collision cost, lateral cost denotes the horizontal distance of local trajectory from the obstacle, longitudinal cost denotes the vertical distance of local trajectory from the nearest obstacle. The transition cost limits the ability of vehicles to switch frequently between candidate paths, ensuring smooth progress.

Then, cost normalization is carried out. Each trajectory is evaluated with additional cost function, and three different normalized cost measures are calculated to avoid one cost in the evaluation function occupying too much. Finally, the cost function is used to calculate the cost, and a local path with the lowest cost is selected. The cost function is shown in Equation (15):

$$H(x) = l \cdot \alpha + d \cdot \beta + v \cdot \gamma + k \cdot \delta \tag{15}$$

where $l$ represents the distance between the vehicle and the lane centerline, $d$ represents the distance between the vehicle and the obstacle, $v$ represents the rate of change in speed, and $k$ represents the path curvature.

## 5. Fusion Algorithm and Simulation Experiment

*The Storage Structure of Nodes Is Optimized*

The traditional A* algorithm cannot avoid the random dynamic and static obstacles, and there are security risks. The Open_Planner algorithm tends to fall into local optima when planning local paths between the current node and the next node. To address the above problems, the following improvements were made. The key nodes on the path planned by the improved A* algorithm were used as the input nodes of the Open_Planner

algorithm in local planning, and, finally, the path planning was completed by the fusion algorithm. The fusion algorithm flow is shown in Figure 6.

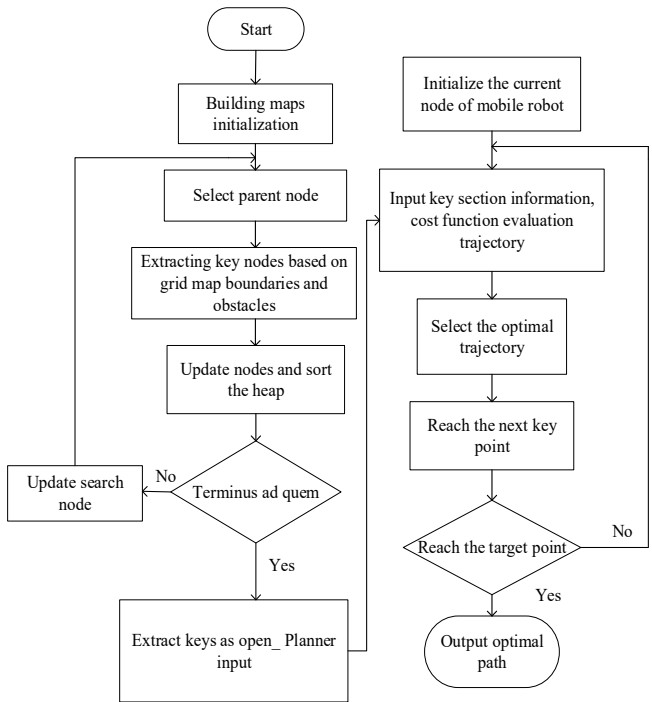

**Figure 6.** Flow chart of fusion algorithm.

In Figure 6, firstly, in the global static map, A* algorithm is used to plan a global path, obtain the key points in the A* algorithm, delete the redundant extended nodes, put the key points into the list, and use heap sort to rearrange the node order. The key points are used as target points in the local planning of the Open_Planner algorithm. Key sections and position information are input, and multiple trajectories are obtained by sampling the intercepted global paths. Three cost functions, priority cost, collision cost and transition cost, are used to evaluate the sampling path. Finally, an optimal trajectory is selected as the path to reach the next target point, and the above steps are repeated until the final target point is reached to realize the fusion of the two improved algorithms.

## 6. Experimental Results

### 6.1. Simulation Experiment of Improved Algorithm Based on Key Points

The simulation experiments of the traditional A* algorithm and improved A* algorithm were carried out using Matlab. The raster map sizes were set to 20 × 20 and 30 × 30. The path-planning experiments were carried out using two algorithms in two maps, respectively. The path planning of the traditional A* algorithm is shown in Figure 7a,b, and the path planning of the improved A* algorithm is shown in Figure 7c,d. The red number in the figure represents the number of nodes in the path, the blue solid line is the planned route, the black grid is the obstacle, and the white grid is the driving area.

Figure 7a,b shows the simulation results of the conventional A* algorithm. From the experimental results, it can be seen that although the complete path planning can be completed from the starting point by following the flow of the A* algorithm, there are many intermediate nodes, which affects the planning efficiency. Figure 7c,d shows the experimental results of improving the A* algorithm using key points. Long-distance linear path planning is achieved by connecting two key points by removing unnecessary expansion points in the middle, and then the nodes are searched with the heap sorting optimization algorithm. Simulation results show that the improved A* algorithm improves the efficiency of the planning extension nodes by 71.5%. From Figure 8, it can be seen

that the number of turning points is reduced by 46.2%. It can be seen that the improved algorithm not only improves the planning efficiency, but also plans fewer path nodes and is more efficient compared with the traditional algorithm.

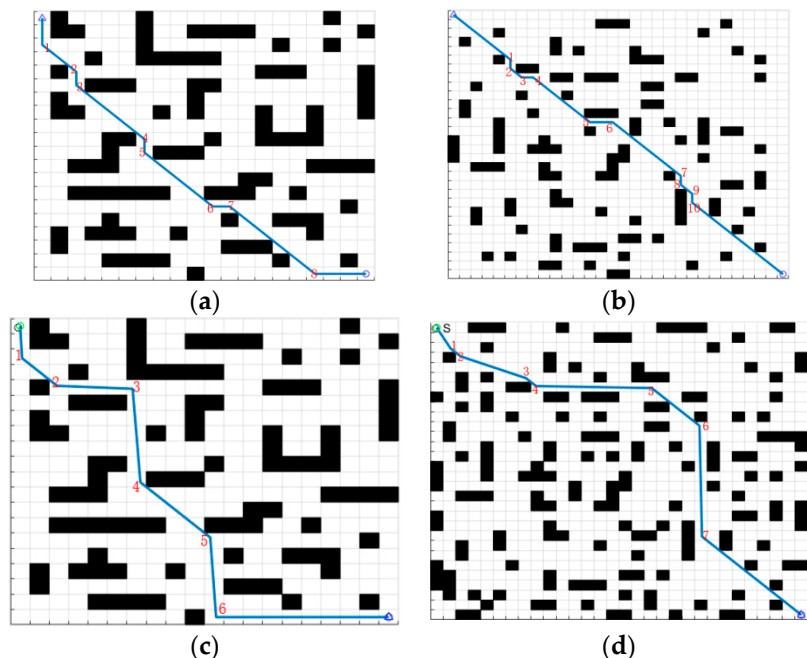

**Figure 7.** Traditional A* algorithm and improved A* algorithm. (**a**) Case 1, (**b**) Case 2, (**c**) Case 1 (**d**) Case 2. (The red number in the figure represents the number of nodes in the path, the blue solid line is the planned route, the black grid is the obstacle, and the white grid is the driving area).

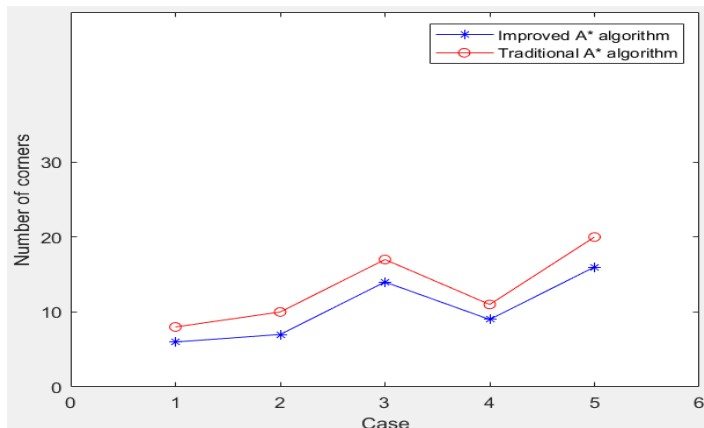

**Figure 8.** Comparison of the number of nodes.

### 6.2. Path-Smoothing Experiments

There are many turning points on the routes in the park. If the path at the turning point is not smooth enough, it will not only affect the vehicle turning efficiency, but also may lead to path-planning failure. Therefore, this paper adopted minimum snap to optimize the trajectory at the turning point and obtain the continuity curve between two points using time, velocity and acceleration constraints. This makes the route smoother.

Figure 9 shows the original path and the path after smoothing. From the figure, it can be seen that after the path processing, the smoothness of the path at the corners is obviously optimized, and the path planning can be carried out better in an environment containing a large number of corners.

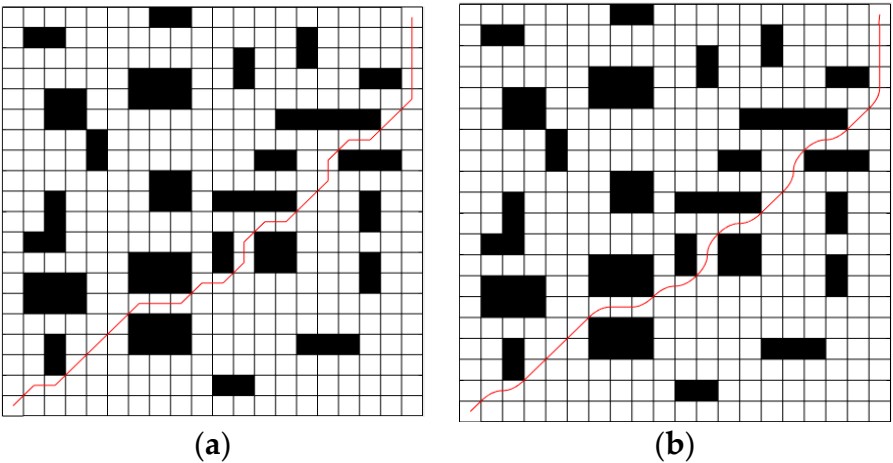

**Figure 9.** Path-smoothing experiments. (**a**) traditional algorithms, (**b**) smooth processing path.

*6.3. A-OP Algorithm Simulation Experiment*

The fusion algorithm designed in this paper enables the mobile robot not only to complete global path planning, but also to achieve local planning to avoid obstacles. The fusion algorithm was simulated in $20 \times 20$ and $40 \times 40$ maps. Static and dynamic obstacles were added to the map, and the simulation results of the fusion algorithm are shown in Figure 10. Figure 10a,b shows the experiments in different environments. The black dashed line is the planned path of the improved A* algorithm, and the blue solid line is the planned path of the fusion algorithm.

The simulation results of the fusion algorithm in Figure 10a,b show that the fusion algorithm starts from the starting point and there are no obstacles on the path until the first critical point. Figure 11 shows the state of the simulated vehicle. It can be seen that the path planned by the fusion algorithm is close to the path planned by the improved A* algorithm and conforms to global path planning. When there are obstacles between two critical points, the locally planned route will avoid the obstacles until it reaches the target point. The experimental results in Table 1 show that in the $20 \times 20$ map, the fusion algorithm uses the feature points in the A* algorithm as intermediate target points when planning the path, compared to the improved A* algorithm which completed regularized path planning from the start point to the target point. The experimental data in Table 1 shows that the fusion algorithm improves the planning efficiency by about 61.09% compared to the traditional A* algorithm, the fusion algorithm avoids the influence of dynamic and static obstacles on robot motion, and the planned path can avoid the obstacles in time to achieve safety. Likewise, the fusion algorithm achieves ideal obstacle avoidance in complex environments with many obstacles.

**Table 1.** Comparison of experimental results of the two algorithms.

| Algorithm | Map | Path Length | Time/s |
| --- | --- | --- | --- |
| Traditional A* | Simple map | 31.38 | 10.97 |
| Fusion algorithms | Simple map | 30.88 | 7.88 |
| Traditional A* | Complex map | 95.43 | 20.33 |
| Fusion algorithms | Complex map | 94.65 | 15.63 |

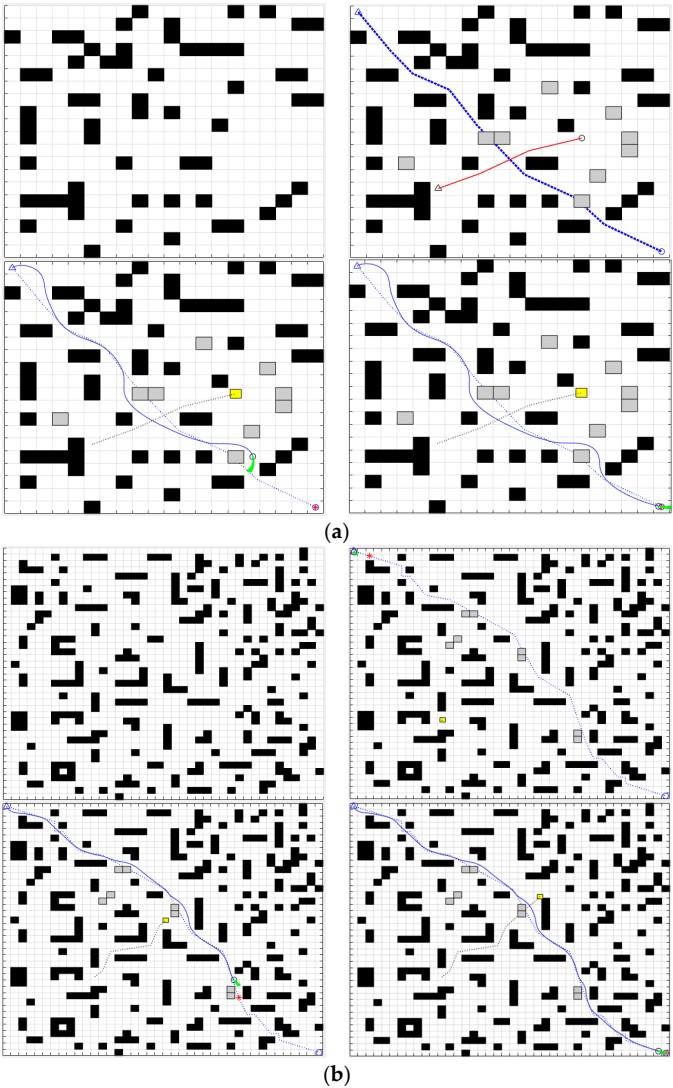

**Figure 10.** Simulation experiments of A-OP fusion algorithm. (**a**) simple environment, (**b**) complex environments.

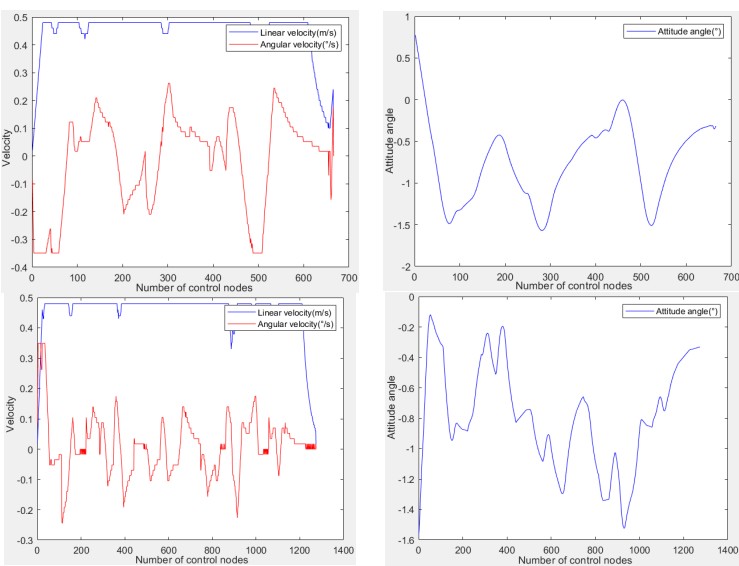

**Figure 11.** Simulated vehicle status diagram.

### 6.4. Real-Time Motion Planning

In order to verify whether the fusion algorithm can be applied in practice, the fusion algorithm was embedded in an unmanned sweeper vehicle for validation. The unmanned sweeper was built on the Autoware development platform with integrated sensors such as LiDAR and IMU. LiDAR is used to scan environmental information and generate highly accurate maps of the environment. The unmanned sweeper used for the experimental validation is shown in Figure 12, and the motion parameters were set as shown in Table 2.

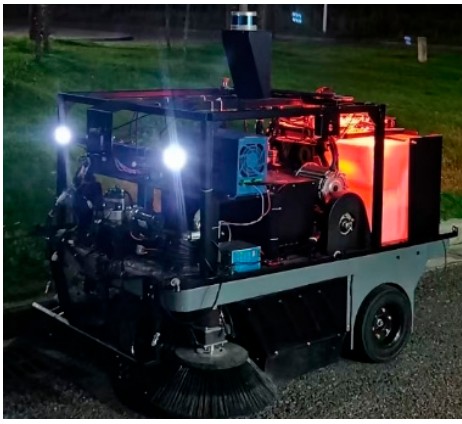

**Figure 12.** Unmanned sweeper for the experiment.

**Table 2.** Movement parameters of unmanned sweeper.

| Parameters of Unmanned Sweeper | Numeric Value |
| --- | --- |
| Maximum line speed (km/h) | 2 |
| Maximum angular velocity (°/s) | 0.35 |

The roslaunch work_driver work_driver. launch command was run to open the chassis driver node, and then the roslaunch work work_keyboard run. Through the I, J, K, L four direction keys, the movement of the vehicle was controlled. Then, the ntd_mapping algorithm was used to construct the environment map. After the construction, the global map was saved as pcd map, and the final high-precision vector map was created according to the pcd map.

### 6.5. Experimental Verification of Improved A* Algorithm and Fusion Algorithm

Two random obstacles (white board and car) were added to the experimental environment, the starting point and the ending point were set on the same map, and the improved A* algorithm and fusion algorithm were, respectively, used to plan the path. The experimental map is shown in Figure 13.

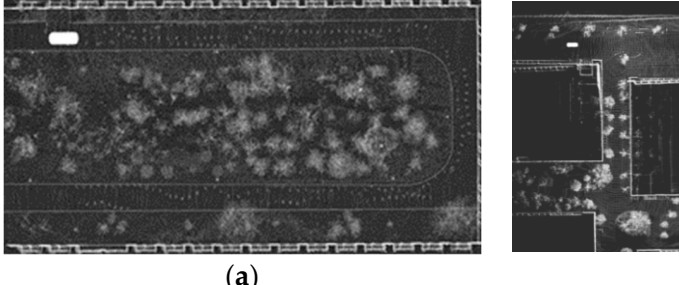

**Figure 13.** Environment map. (**a**) Simple environment map (**b**) Complex environment map.

The solid blue line in Figure 14 indicates the global route planned by the improved A* algorithm. Figure 15 shows the effectiveness of the fusion algorithm for path planning and actual vehicle avoidance for an unmanned sweeper. The improved A* algorithm plans a path with better smoothness and planning time than the traditional A* algorithm, but the unmanned sweeper vehicle cannot avoid obstacles on the road. The fusion algorithm performs local planning on the basis of the global path. When an obstacle is detected ahead, the driverless sweeper performs an early obstacle avoidance maneuver and obstacle-avoidance path planning. Then, when the next obstacle is detected, it also performs an early obstacle-avoidance maneuver and changes its path until it reaches the target point. The experimental results are shown in Table 3. Compared to the improved A* and Dijkstra algorithms, the fusion algorithm outperforms both in terms of time and planned path length because it is able to plan ahead for static and dynamic obstacles in the path (Figure 16), reducing path wastage due to replanning. The fusion algorithm improves the planning efficiency by 65.46% in practice. It can be seen that the fusion algorithm can meet the requirements of optimal path planning and the car can reach its final destination efficiently and safely. Dijkstra's algorithm produces replanning behavior when planning paths with known static and dynamic obstacles, which often leads to additional path loss. However, our proposed method enables early path planning in regions with obstacles through the evaluation function of the obstacle factor, and the reduction in the number of replans reduces unnecessary losses and, subsequently, relatively shortens the planned path. It is worth raising the point that, this study was carried out with undirected graphs and containing unknown obstacles, which may not be fair to the Dijkstra and A* algorithms; thus, future work is needed to add more experimental comparison groups to validate the soundness of the algorithms.

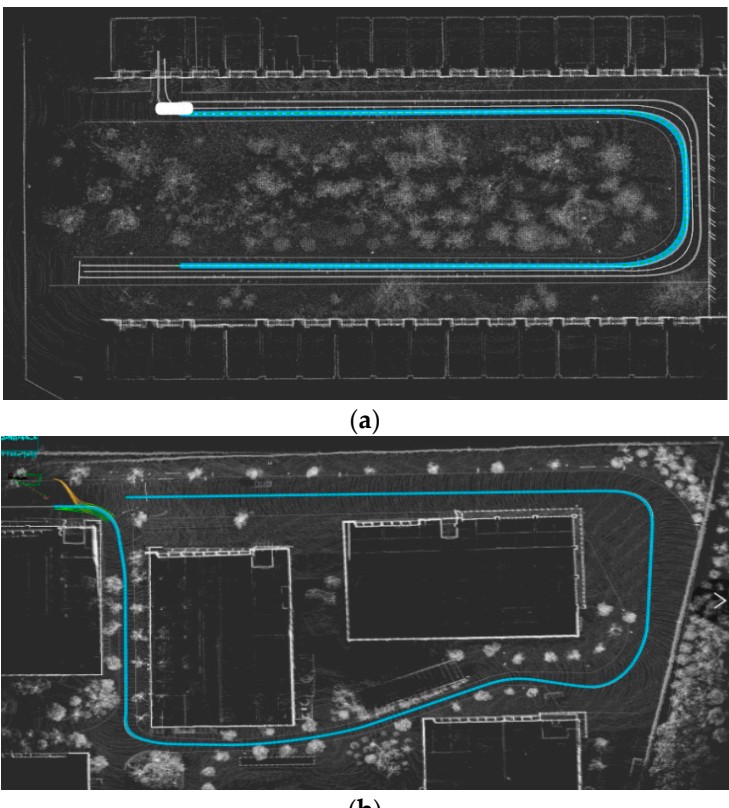

(**a**)

(**b**)

**Figure 14.** Global planning route. (**a**) Simple environment map; (**b**) complex environment map.

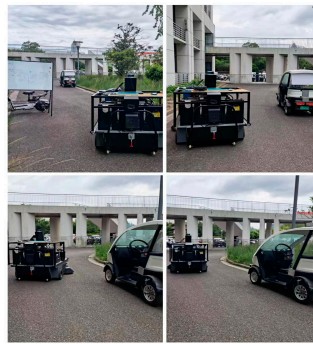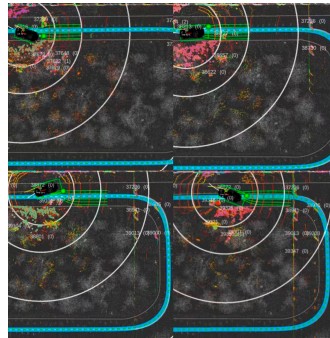

**Figure 15.** Obstacle avoidance experiments.

**Table 3.** Comparison of experimental results of the three algorithms.

| Experimental Number | Dijkstra | | Improved A* | | Proposed Method (A-OP) | |
|---|---|---|---|---|---|---|
| | Path Length/m | Time/s | Path Length/m | Time/s | Path Length/m | Time/s |
| 1 | 11.45 | 6.37 | 10.13 | 4.22 | 9.71 | 1.37 |
| 2 | 56.87 | 22.09 | 54.28 | 19.91 | 53.53 | 15.18 |
| 3 | 33.15 | 14.99 | 31.43 | 13.22 | 30.93 | 9.91 |
| 4 | 32.17 | 12.01 | 29.66 | 10.87 | 29.04 | 6.67 |

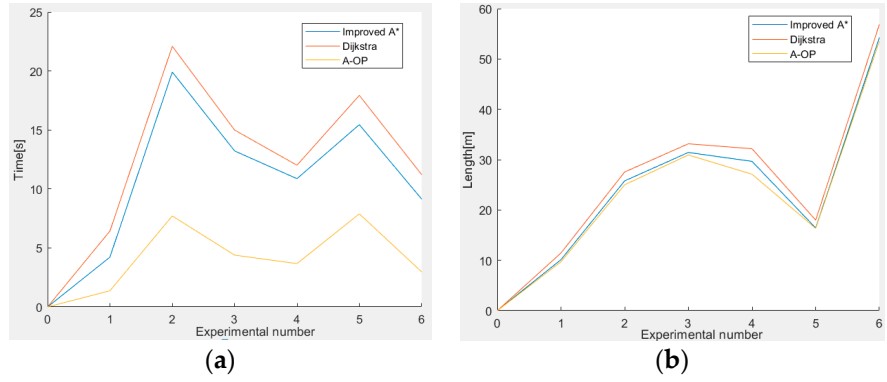

**Figure 16.** Algorithm performance. (**a**) Planning time, (**b**) Path length.

## 7. Conclusions

The traditional A* algorithm has some shortcomings in global path planning, such as low efficiency, many redundant points, and uneven path, etc., and it cannot realize the function of avoiding obstacles. From the point of view of path-planning efficiency and obstacle-avoidance function, this paper designed an optimization algorithm based on the fusion of improved A* and Open_Planner. Based on the global optimal path, dynamic obstacle avoidance was carried out to ensure that the target point can be reached efficiently and safely. In this paper, the selection of the nodes of the A* algorithm, the removal of unnecessary extension points and the improvement of node storage structure improve the computational efficiency of A* algorithm. The key points of the global planning are taken as the intermediate nodes of the local-planning algorithm, and, finally, the algorithm fusion was realized. The simulation results show that the efficiency of the A-OP algorithm is significantly improved, and the removal of redundant points in the path means the path tends to be smoother.

The A-OP algorithm not only makes up for the defects of the A* algorithm, and completes the global path planning, but also avoids the obstacles in the path in time, and reaches the destination safely and efficiently. In subsequent research, the algorithm will

continue to be optimized to continuously improve the actual efficiency and application value of path planning.

**Author Contributions:** Conceptualization, Y.M. and P.P.; methodology, Q.S.; software, Y.M.; validation, Y.M.; formal analysis, Y.M.; investigation, P.P.; resources, P.P.; data curation, Q.S.; writing—original draft preparation, Q.S.; writing—review and editing, P.P.; visualization, Y.M.; supervision, Q.S.; project administration, Q.S.; funding acquisition, P.P. and Q.S. All authors have read and agreed to the published version of the manuscript.

**Funding:** National Natural Science Foundation of China (52202496, 61771265); Postgraduate Research & Practice Innovation Program of School of Information Science and Technology, Nantong University (NTUSISTPR21-010); the Natural Science Foundation for Colleges and Universities in Jiangsu Province (22KJB520007) and the Science and Technology Foundation of Nantong (MS22021034, MS12022015).

**Institutional Review Board Statement:** Not applicable.

**Informed Consent Statement:** Not applicable.

**Data Availability Statement:** Not applicable.

**Conflicts of Interest:** The authors declare no conflict of interest.

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
