# Peer review of "Research on the Path Planning of Unmanned Sweepers Based on a Fusion Algorithm"

_applsci, doi:10.3390/app13042725_

Round 1
Reviewer 1 Report
The presented method is based on the well-known A* algorithm, considered a global planning algorithm with significant drawbacks, including lack of smoothness and inefficient way of solving the planning problem. The authors provide a considerably good overview of the existing path-planning algorithms, which justifies the use of the proposed method.
The main findings are based on simulated and real-life experiments, enabling numerical comparison of evaluating the method's efficiency and visual assessment of the trajectory smoothness in static and "dynamic" environments.
While the results are scientifically interesting and sound, some minor drawbacks should be noticed and corrected before publishing:
1) In the introduction chapter, the authors refer to different existing scientific works and reports. However, in different contexts, a "etc.." is used. It should be considered as a weak reference style stating that there are other results, but they are not mentioned. So, whether they are irrelevant should not be referred to or vice versa.
2) The authors refer to ANN-based path planning methods, but the reference is not provided. The reference has to be provided.
3) The methods explaining part provide several equations, which are the essence of the report. Unfortunately, the used indexes and some variables are not described clearly, making it hard to read.
4) It is not clear how the time intervals T are selected, the criteria for selecting the interval length and the number of intervals.
Another minor remark is on A* as such - it should be considered a generic algorithm for optimal path planning tailored for particular necessities. So, comparing other algorithms to generic ones is not a good practice.
Author Response
We carefully read and revised the paper according to the reviewers' comments, and the specific responses to the reviewers' comments are included in the attachment.

Reviewer 2 Report
I have attached my review.

Author Response
We have carefully read and revised our paper according to the comments of the reviewers, and we have put the specific replies to the reviewers' comments in the annex.

Reviewer 3 Report
Motion planning is one of the most important challenges in robotics. The method presents a method that combines the traditional A* with another motion planning method called open_planner. The authors include experimentation in simulation as well as with a real robotic platform.
My concern about the method is that there is no sufficient information to validate the method.
- There is no reference or background about the called open_planner. Is this an algorithm or a library?
- Experiments are confusing. It has been demonstrated that Dikstra algorithms guarantee optimal trajectories. However, the presented method shows better performance. This is not possible unless the methods are compared unfairly.
- If you are planning continuous trajectories, then you should compare your method against similar methods such as RRT* or SST*
- Labels and better explanations of the figures are required.
Author Response

(The authors gave the same response as above.)

Round 2
Reviewer 3 Report
I still see the already described flaws
Second review: Since Dijkstra's algorithm is optimal, it gets the shortest path, in addition A* also gets optimal paths. However, in table 3 their method gets 9.71 (no units have been specified) while Dijkstra gets 11.45. The same behaviour is observed in table 1 for A*. This is not possible in a fair experimentation.
It might be that the authors are improving a secondary method such as sorting the nodes. However, this is not going to change the path distance.
Table 3 is neither called in the text nor explained. So, their results are not supported.
Figure 11 still does not have named axis.
Author Response

(The authors gave the same response as above.)
